# Racial and ethnic disparities in benefits eligibility and spending among adults on the autism spectrum: A cohort study using the Medicare Medicaid Linked Enrollees Analytic Data Source

Teal W. Benevides[1]*, Henry J. Carretta[2], George Rust[2], Lindsay Shea[3]

1 Department of Occupational Therapy, College of Allied Health Sciences, Augusta University, Augusta, GA, United States of America, 2 College of Medicine, Florida State University, Tallahassee, FL, United States of America, 3 A.J. Drexel Autism Institute, Drexel University, Philadelphia, PA, United States of America

* tbenevides@augusta.edu

**Data Availability Statement:** The data used and described in this study are not publicly available, and the authors legally cannot make this data

## Abstract

### Background

Research on children and youth on the autism spectrum reveal racial and ethnic disparities in access to healthcare and utilization, but there is less research to understand how disparities persist as autistic adults age. We need to understand racial-ethnic inequities in obtaining eligibility for Medicare and/or Medicaid coverage, as well as inequities in spending for autistic enrollees under these public programs.

### Methods

We conducted a cross-sectional cohort study of U.S. publicly-insured adults on the autism spectrum using 2012 Medicare-Medicaid Linked Enrollee Analytic Data Source ($n$ = 172,071). We evaluated differences in race-ethnicity by eligibility (Medicare-only, Medicaid-only, Dual-Eligible) and spending.

### Findings

The majority of white adults (49.87%) were full-dual eligible for both Medicare and Medicaid. In contrast, only 37.53% of Black, 34.65% Asian/Pacific Islander, and 35.94% of Hispanic beneficiaries were full-dual eligible for Medicare and Medicare, with most only eligible for state-funded Medicaid. Adjusted logistic models controlling for gender, intellectual disability status, costly chronic condition, rural status, county median income, and geographic region of residence revealed that Black beneficiaries were significantly less likely than white beneficiaries to be dual-eligible across all ages. Across these three beneficiary types, total spending exceeded $10 billion. Annual total expenditures median expenditures for full-dual and Medicaid-only eligible beneficiaries were higher among white beneficiaries as compared with Black beneficiaries.

available due to a Data Use Agreement with Centers for Medicare and Medicaid (CMS) that prohibits data sharing (DUA#: RSCH-2020-55304). The data are available for purchase from CMS following a data use request from the third-party vendor, the Research Data Assistance Center (ResDAC). Researchers seeking to purchase data should visit www.resdac.org for instructions, guidance, and costs of CMS data. The specific CMS data used in this study were extracted by a third-party vendor from the Medicare-Medicaid Enrolled Linked Data Source, 2012 files.

**Funding:** This research has been supported in full with an AOTF Health Services Research grant (Grant# AOTF2019HSRBenevides) funded by the American Occupational Therapy Foundation (Authors TB, HC, GR). Website: www.aotf.org The funders had no role in study design, data collection and analysis, decision to publish, or preparation of the manuscript.

**Competing interests:** The authors have declared that no competing interests exist.

**Abbreviations:** ASD, Autism spectrum disorder; CMS, Centers for Medicare and Medicaid; DI, disability insurance; ID, Intellectual disability; MMLEADS, Medicare-Medicaid Linked Enrollees Analytic Data Source; OASI, old age and survivors insurance; ESRD, end-stage renal disease; SSA, Social Security Administration; SSDI, Social Security Disability Income; SSI, Social Security Income.

## Conclusions

Public health insurance in the U.S. including Medicare and Medicaid aim to reduce inequities in access to healthcare that might exist due to disability, income, or old age. In contrast to these ideals, our study reveals that racial-ethnic minority autistic adults who were eligible for public insurance across all U.S. states in 2012 experience disparities in eligibility for specific programs and spending. We call for further evaluation of system supports that promote clear pathways to disability and public health insurance among those with lifelong developmental disabilities.

## Introduction

Extensive research on children and youth with autism spectrum disorder (ASD) reveal racial and ethnic disparities in a variety of health services and access to care [1–3]. For example, Black youth on the autism spectrum are significantly less likely to be identified or diagnosed early in life, as compared to white children [2, 4, 5]. Racial and ethnic disparities persist in minority children's ability to obtain high-quality medical services including specialty care, therapy services, and educational services [6–12]. Although an increasing number of youth on the autism spectrum are aging into adult systems of care, and autism is a lifelong developmental disability [13, 14], parallel research to understand how these disparities persist as individuals age is understudied [15, 16].

The "support cliff" facing transition-age youth with ASD has been well documented [14, 17] using population-based samples as impacting access to community-based services, postsecondary education opportunities, and health outcomes. Poor physical and mental health outcomes among autistic adults, in particular, have been a focus of much research, and greater utilization of high-cost healthcare such as emergency department utilization are concerning [18–21].The first step in understanding pathways for these suboptimal outcomes is to identify the mechanisms which autistic adults rely upon for health benefits. Since most U.S. healthcare coverage for adults occurs through employer-based health insurance, autistic adults are less likely than their counterparts with other diagnoses to be employed [22, 23]. Population-based estimates suggest that only 55% of autistic adults are employed within 6 years of high-school graduation. Without employer-based health insurance, continuous access to healthcare coverage that addresses physical and mental health needs of adults on the autism spectrum requires eligibility for, and enrollment in, public health insurance such as Medicare and/or Medicaid. Understanding racial-ethnic disparities in public insurance enrollment is a critical step for identifying ways to eliminate disparities in healthcare services and health outcomes in the adult ASD population.

If individuals are unable to either obtain healthcare benefits through employment or individually in the private marketplace, they will be uninsured unless they can obtain coverage through public-sector programs such as Medicaid or Medicare. Eligibility for public health care coverage for adults via Medicare and Medicaid are available for disabled individuals who meet criteria for those benefits, for individuals who meet income and asset requirements (Medicaid only), or as an entitlement program based on age (Medicare only). Both Medicare and Medicaid are considered safety net programs for unemployed or under-employed individuals with disabilities, including those with autism spectrum disorder. However, these systems are difficult to navigate and understand. For example, obtaining a Social Security Disability determination requires claimants to present their case supporting a disability; frequently the

initial decisions are denial and require additional application [24]. Recently, Boyd and Rutkowski [25] hypothesized that disability decisions for more 'subjective' diagnoses, such as depression, are influenced by race and gender (p. 8). Given that ASD diagnostic racial and ethnic disparities persist, the burden of obtaining such documentation to support *adult* disability determination for a diagnosis of ASD (typically diagnosed in childhood), and which is based on clinical judgement, disproportionately affects individuals of minority groups who already have difficulty obtaining diagnoses.

Although the research documenting poor health outcomes among autistic adults is pervasive, the samples contributing to that data primarily include privately-insured autistic individuals [18, 26]. Additionally, those samples tend to reflect primarily white, non-minority samples, and therefore are not ideal for examining racial and ethnic differences in healthcare access and use. Little research has been conducted using national samples of publicly insured adults on the autism spectrum for whom benefits are available, although some research exists on Medicare-only samples [27, 28]. This study fills this research gap by using a national sample of Medicare-Medicaid enrollees across all 50 states and the District of Columbia to evaluate racial and ethnic differences in public health coverage enrollment for autistic adults. We also analyzed the racial and ethnic differences that might exist in out-of-pocket and program spending for medical care among Medicare and/or Medicaid-enrolled adults with ASD that suggest disparities in utilization.

## Methods

The study received institutional review board approval as human subjects research by Augusta University IRB Committee A for project 1413487–4, with a HIPAA and consent waiver to conduct the analyses due to lack of ability to consent persons with protected health information (PHI) in health record data from Centers for Medicare and Medicaid.

### Data source

We used the 2012 Medicare-Medicaid Linked Enrollees Analytic Data Source (MMLEADS) V2 research-identifiable files for primary data analyses (see the 'S1 Data' for information about how to access this data), representing the latest data release for this source. The 2012 MMLEADS file comprises 60,043,743 individual beneficiaries with coverage from Medicare, Medicaid, or who are dually-enrolled by both Medicare and Medicaid. The MMLEADS 2012 data are considered research-identifiable data with PHI. The MMLEADS V2 files are prepared as four separate data files; two are person-level files (Beneficiary, Condition files) and two are service-level files (Medicare Services, Medicaid Services). In order to accomplish the primary aims, we used the person-level files in the reported analyses below.

### Sample identification

Our sample frame included any beneficiary who met Chronic Condition Warehouse criteria for "Autism Spectrum Disorder" (ASD), "Developmental Disability", or "Intellectual Disability" (ID) Medicare or Medicaid claims records in 2009, 2010, 2011, or 2012. With that sampling frame, we purchased data for those eligible beneficiaries in the 2012 MMLEADS data release. See S1 Table for criteria used for these conditions by CMS and other methodological considerations for this data source. These beneficiaries comprised a possible analytic sample of $n = 1,610,364$ unique individuals of any age with either ASD, ID, and/or developmental disability who were eligible for either Medicare, Medicaid, or were dually-eligible in the 2012 year. This sample represents approximately 2.68% of the total 2012 MMLEADS V2 beneficiaries available.

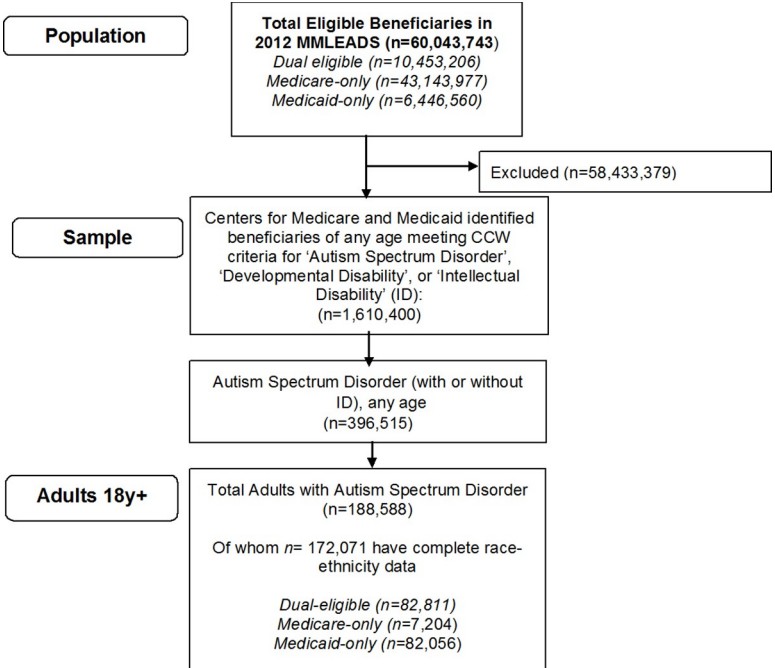

**Fig 1. Identification of 2012 MMLEADSv2 analytic sample of adults with autism spectrum disorder.**

For this analysis, we were interested in those with autism spectrum disorder, with or without intellectual disability, who either currently or ever met criteria for ASD (S1 Table). Our research question in this study was focused on eligibility for benefits among adults, and therefore, we excluded children younger than 18 years from our analyses (Fig 1).

### Dependent variables

A full description of all variables included in this study is available in the MMLEADSv2.0 User Guide [29]; a summary is provided here. The primary eligibility dependent variable was beneficiary type, defined as Medicare-Medicaid program eligibility. This variable comprised a mutually-exclusive variable that documented a beneficiary's annual status of Medicare-only, Medicaid-only, or dual-eligible for Medicare and Medicaid (categories of partial-dual, 'qualified Medicare beneficiary'[QMB-only] dual, or full-dual). The annual eligibility variable was created by CMS using a published algorithm that identifies any months where the beneficiary had both Medicare and Medicaid coverage; having at least one month with dual coverage resulted in being classified as one of the dual types. Because the majority of the dual-eligible autism sample was identified as *full-dual* eligible and not QMB-dual or partial-dual, the models and statistical analyses include the three primary eligibility types: Medicaid-only, Medicare-only, and full-dual beneficiaries. Over 88% of full-dual beneficiaries had 12-months of this coverage type; 93% of Medicaid-only beneficiaries were covered for 12-months; and 92% of Medicare-only beneficiaries were covered for 12-months. We explicitly did not restrict to 12-months of required coverage, since this would exclude people who died during the year, and people who became eligible for dual-status at some point in the year. Since the purpose was to understand factors that contribute to eligibility status (not specific person-month utilization), this intended sample reflects the research purpose.

For those with *Medicare* benefits, we describe several other eligibility variables. We were primarily interested in the original reason for Medicare entitlement, which included: old age

and survivors insurance (OASI); disability insurance (DI); end-stage renal disease (ESRD); or a combination of DI and ESRD.

For *Medicaid*-eligible beneficiaries, we were interested in the basis for eligibility from the beneficiary's most recent month of Medicaid eligibility. The most recent month of eligibility was the same as the modal basis of eligibility in 97% of beneficiaries. We also report whether benefit was offered through any waiver program based on the first month of the year. The waiver of the first month of the year was the same as the modal waiver type across all 12 months in 95% of beneficiaries. To preserve confidentiality, we suppressed reporting of some waiver eligibility categories in accordance with our data use agreement with CMS. The following waiver programs had too few beneficiaries to report: 1915(C) HIV/AIDS,1915(C) serious mental illness, 1915(C) medically fragile, 1115 HIFA, and 1115 family planning.

Our secondary dependent variable was annual spending for services, both from the payer (system) perspective and the beneficiary (out-of-pocket) perspective. Spending variables were available in the 2012 MMLEADSv2.0 Beneficiary file, and included aggregated CMS spending which reflect unduplicated costs across Medicare and Medicaid. These variables are quantified as costs by eligibility type (e.g. Medicaid-only, Medicare-only, and full-dual eligible). Spending variables were obtained from fee-for-service (FFS) claims in either Medicare ($n = 72,076$) and/ or Medicaid FFS ($n = 53,307$). Medicare spending variables included Medicare-paid Part A and Part B services. If the beneficiary was a Medicare Managed Care (Part C) beneficiary, only the beneficiary premiums and spending for Part D drugs were available for inclusion in the spending totals, because encounter claims for Part C were not reported to CMS until 2018 [29]. Those with Medicare Part D drug coverage were included. Medicaid spending variables included both the state paid and federal share of payments for services, as well as beneficiary deductible and coinsurance amounts. The Medicaid source data were MAX claims. We used four annual spending variables: annual Medicare payments for services (total paid by Medicare in 2012), annual Medicaid payments for services (total annual Medicaid payments amounts for all services including premiums for care), annual Medicare beneficiary payments (beneficiary responsibility for services including deductibles and co-insurance), and annual Medicaid beneficiary payments (responsibility for co-insurance or deductibles). We created two total spending variables which reflected a sum of payments by Medicare and payments by the beneficiary for Medicare services in the 2012 claim year, and a sum of payments by Medicaid and payments by the beneficiary for Medicaid services in the 2012 claim year.

## Independent variable

Race and ethnicity comprised our primary independent variable. Because our analytic data source contained two separate race-ethnicity variables, one from Medicare and one from Medicaid, we chose to rely on the Medicare variable as the original source of race and ethnicity. This decision was informed by research that evaluated the validity of the Medicare Research Triangle Institute (RTI) variable in comparison to self-reported race and ethnicity [30]. The Medicare RTI race variable has been shown to have both high sensitivity and specificity with self-reported race and ethnicity for white, Black, Hispanic, and Asian/Native Hawaiian/Pacific Islander categories [30]. For 52% of our ASD sample without a Medicare race code, we relied on the Medicaid race-ethnicity variable, which comes from the MAX files as reported by states. Of our adult ASD sample of $n = 188,588$ beneficiaries, $n = 16,517$ (8.76%) had 'unknown' race or ethnicity on both the Medicare and Medicaid race-ethnicity variables, and these beneficiaries were coded as 'missing' and deleted listwise in the analyses focused on race-ethnicity. The original race-ethnicity categories included: white ($n = 120,276$), Black/African-American ($n = 31,341$), Asian/Pacific Islander ($n = 4,291$), American Indian/Alaska Native ($n = 1,306$),

Other/More than one ($n = 1,143$), and Hispanic ($n = 13,714$). Because cell sizes were low for some analyses, and reporting of cell sizes 11 or fewer beneficiaries is prevented by our Data Use Agreement, we collapsed groups into white, Black, Hispanic, Asian/Pacific Islander, and Other/More than One (which included beneficiaries of Alaska Native, American Indian, those reporting more than one race-ethnicity, and beneficiaries identified as 'other').

White, Black, Hispanic, Asian/Pacific Islander, and Other/More than One (which included beneficiaries of Alaska Native, American Indian, those reporting more than one race-ethnicity, and beneficiaries identified as 'other').

## Covariates

We included age, sex, intellectual disability, an indicator variable for costly chronic conditions, county median income, rural county status and Census Bureau geographic region of residence as covariates. Most variables were drawn from the MMLEADS Beneficiary or Condition file. A costly chronic condition indicator was created from variables provided in the Conditions MMLEADS data source. We used the Centers for Disease Control and Prevention to identify the most costly and economically impactful conditions to include in this indicator variable [31], which included: depression; epilepsy; diabetes; rheumatoid and osteoarthritis; heart disease conditions such as acute myocardial infarction, hypertension, stroke, congestive heart failure; cancers including breast, colorectal, lung, and prostrate; Alzheimer's disease; and obesity. This variable was coded as '1, Yes' if the beneficiary currently had one of thirteen chronic conditions, according to the Chronic Condition Warehouse algorithm [29]. We included county median income, drawn from the 2010 Census records and merged by FIPS county code with beneficiary data. Sex was dichotomous (male, female). Rural status was a dichotomous indicator (non-rural, rural county) based on the Health Resources and Services Administration Federal Office of Rural Health classification of counties using county and state FIPS code for linking. Non-rural counties were 'large central metro', 'large fringe metro', 'medium metro' and 'small metro'. Rural counties were 'micropolitan' and 'non-core' counties. Approximately 1% of the sample was missing data that would allow rural classification ($n = 1,992$). U.S. Census geographic region was based on four categories (Northeast, South, Midwest, and West), using state of residence from the Beneficiary file. A dichotomous indicator of intellectual disability (yes/no) was used as a covariate since intellectual disability is frequently a reason for increased services and spending [32].

## Data analysis

We used bivariate chi-square tests of independence to evaluate differences in beneficiary type by race-ethnicity. With five race-ethnicity groups and some categorical variables with up to five categories ($u = 16$), we exceeded the sample size needed of $n = 3,733$ to detect small effects ($w = .10$) at 99% power using chi-square [33]. We originally planned to use multinomial logistic regression to evaluate the relationship of race-ethnicity and three categories of beneficiary status, however, specification errors due to the small sample of Medicare-only beneficiaries limited by age and race-ethnicity prevented accurate estimation with all three predicted eligibility types. We therefore used logistic regression models to predict Full-dual status (1) versus Medicaid-only status (0) by race-ethnicity in using five age categories up to age 64 years, while controlling for gender, intellectual disability, costly chronic condition indicator, county median income, rural county status, and geographic region of residence. Models were examined for specification errors with the link test, and variables were examined for linearity and large standard errors within variables. Post-estimation marginal effects with 95% confidence intervals predicting the probability of being "Full-Dual eligible" for each race-ethnicity group,

while holding other covariates constant are presented. Unadjusted median spending with 95% confidence intervals by Medicare, Medicaid, and by beneficiaries are reported by race-ethnicity. In all analyses, reporting in accordance with CMS data use agreement required suppression of cell values and percentages when 11 or fewer counts were present. Statistically significant relationships were identified when $p < .05$.

## Results

### Demographic characteristics

Demographic characteristics by race-ethnicity are reported in Table 1. The majority of the adult sample was white (69.90%, $n = 120,276$), followed by 18.21% Black ($n = 31,341$), 2.49% Asian or Pacific Islander ($n = 4,291$), 1.42% other/more than one race ($n = 2,449$), and 7.97% Hispanic ($n = 13,714$). The majority of the adult sample was between the ages of 23 and 40 years (45.19%), were male (72.48%), and had an intellectual disability (68.15%). Black (OR = 1.60, 95% confidence interval [CI] 1.55–1.64) and Hispanic (OR = 1.17, 95%CI = 1.13–1.22) beneficiaries were significantly more likely to have a co-occurring intellectual disability than white beneficiaries. Black (OR = 0.87, 95%CI = 0.84–0.89), Asian/Pacific Islander (OR = 0.59, 95%CI = 0.55–0.63), and Hispanic beneficiaries (OR = 0.83, 95%CI = 0.80–0.86) were significantly less likely than white beneficiaries to have a costly chronic condition.

**Table 1. Demographic and beneficiary type by race-ethnicity of adults with autism spectrum disorder.**

| | Total ($n$ = 172,071) | | White ($n$ = 120,276) | | Black ($n$ = 31,341) | | Asian/ Pacific Islander ($n$ = 4,291) | | Other/More than One Race ($n$ = 2,449) | | Hispanic ($n$ = 13,714) | |
|---|---|---|---|---|---|---|---|---|---|---|---|---|
| | $f$ | Col % | $f$ | Col % | $f$ | Col % | $f$ | Col % | $f$ | Col % | $f$ | Col % |
| **Male** | 124,714 | 72.48 | 86,811 | 72.18 | 23,021 | 73.45 | 3,161 | 73.67 | 1,769 | 72.23 | 9,952 | 72.57 |
| **Age Category (y)** | | | | | | | | | | | | |
| 18–24 | 58,620 | 34.07 | 37,656 | 31.31 | 12,005 | 38.30 | 1,821 | 42.44 | 898 | 36.67 | 6,240 | 45.50 |
| 25–34 | 47,604 | 27.67 | 31,752 | 26.40 | 9,621 | 30.70 | 1,483 | 34.56 | 671 | 27.40 | 4,077 | 29.73 |
| 35–44 | 23,804 | 13.83 | 17,036 | 14.16 | 4,264 | 13.61 | 544 | 12.68 | 377 | 15.39 | 1,583 | 11.54 |
| 45–54 | 21,751 | 12.64 | 16,953 | 14.10 | 3,230 | 10.31 | 245 | 5.71 | 277 | 11.31 | 1,046 | 7.63 |
| 55–64 | 12,925 | 7.51 | 10,693 | 8.89 | 1,498 | 4.78 | 114 | 2.66 | 146 | 5.96 | 474 | 3.46 |
| 65+ | 7,367 | 4.28 | 6,186 | 5.14 | 723 | 2.31 | 84 | 1.96 | 80 | 3.27 | 294 | 2.14 |
| **Region** | | | | | | | | | | | | |
| Northeast | 41,016 | 23.84 | 30,185 | 25.10 | 5,968 | 19.04 | 727 | 16.94 | 517 | 21.11 | 3,619 | 26.39 |
| Midwest | 47,501 | 27.61 | 36,620 | 30.45 | 8,355 | 26.66 | 663 | 15.45 | 576 | 23.52 | 1,287 | 9.38 |
| South | 50,059 | 29.09 | 31,006 | 25.78 | 14,281 | 45.57 | 697 | 16.24 | 465 | 18.99 | 3,610 | 26.32 |
| West | 32,587 | 18.94 | 21,785 | 18.11 | 2,675 | 8.54 | 2,192 | 51.08 | 878 | 35.85 | 5,057 | 36.87 |
| **Rural County** | 28,390 | 16.50 | 23,381 | 19.44 | 3,343 | 10.67 | 232 | 5.41 | 636 | 25.97 | 798 | 5.82 |
| **Co-Occurring Intellectual Disability** | 117,262 | 68.15 | 79,601 | 66.18 | 23,746 | 75.77 | 2,803 | 65.32 | 1,556 | 63.54 | 9,556 | 69.68 |
| **Current Costly Chronic Condition** | 79,126 | 45.98 | 56,888 | 47.30 | 13,707 | 43.74 | 1,481 | 34.51 | 1,179 | 48.14 | 5,871 | 42.81 |
| **Beneficiary type** | | | | | | | | | | | | |
| Medicaid with disability | 82,056 | 47.69 | 51,663 | 42.95 | 18,378 | 58.64 | 2,642 | 61.57 | 1,048 | 42.79 | 8,325 | 60.70 |
| Medicare-only | 7,204 | 4.19 | 5,951 | 4.95 | 738 | 2.35 | 101 | 2.35 | 117 | 4.78 | 297 | 2.17 |
| Dual-eligible (full-dual) | 79,378 | 46.13 | 59,976 | 49.87 | 11,761 | 37.53 | 1,487 | 34.65 | 1,225 | 50.02 | 4,929 | 35.94 |
| Dual-eligible (partial dual) | 1,573 | 0.91 | 1,254 | 1.04 | 207 | 0.66 | 30 | 0.70 | 30 | 1.22 | 52 | 0.38 |
| Dual-eligible (QMB-only) | 1,860 | 1.08 | 1,432 | 1.19 | 257 | 0.82 | 31 | 0.72 | 29 | 1.18 | 111 | 0.81 |

Data Source: MMLEADS V2, 2012. Centers for Medicare and Medicaid. https://www2.ccwdata.org/documents/10280/19002246/mmleads-user-guide-v2-0.pdf

## Eligibility characteristics

In unadjusted bivariate analyses, we found significant differences in the three primary eligibility groups between race-ethnicity groups ($\chi2(8) = 4000$, p < .001). The majority of white adults (49.87%) were Full-dual eligible for both Medicare and Medicaid. In contrast, only 37.53% of Black, 34.65% Asian/Pacific Islander, and 35.94% of Hispanic beneficiaries were Full-dual eligible for Medicare and Medicare (Fig 2), with the majority only eligible for state-funded Medicaid.

Among *full-dual eligible* adults with ASD, the majority of beneficiaries across all race-ethnicity groups were eligible for Medicaid in the state they live on the basis of 'Blind/Disabled' (*n* = 72,385; 91.19%); a small percentage were eligible on the basis of 'Aged' (*n* = 4,346, 5.48%). Medicaid is operated by states with both federal and state funding; each state has different waiver programs available to beneficiaries. In the first month of 2012, the majority of full-dual eligible beneficiaries (*n* = 34,483, 43.67%) were eligible for the 1915(C) MR/DD waiver. The second most common waiver was 1915(B), with 12.53% (*n* = 9,891), followed by 1115 waivers (*n* = 5,924, 7.50%). Very few (*n* = 237, 0.30%) were eligible for 1915(C) autism waivers. A large percentage were not enrolled in any waiver (*n* = 23,939, 30.32%). Dual eligible beneficiaries may meet different reasons for Medicare entitlement. The majority of full-dual eligible beneficiaries were entitled for Medicare on the basis of Disability Insurance (DI; *n* = 77,616; 97.78%); few individuals across all race-ethnicity groups were originally eligible for Medicare based on OASI (*n* = 1,527; 1.92%), end-stage renal disease (ESRD; *n* = 105; 0.13%), or DI +ESRD (*n* = 130; 0.16%).

Among current *Medicare-only* adults with ASD, the majority of beneficiaries were originally enrolled in Medicare on the basis of DI (*n* = 6,034; 82.86%), with OASI accounting for 16.6% of the Medicare-only beneficiaries (*n* = 1,205). Among those Medicare-only on the basis of OASI, almost 100% were aged 65 or older, suggesting they became eligible on their own age and not that of survivor benefits (sample size and rates suppressed for reporting).

Among current *Medicaid-only* adults with ASD, the majority were eligible on the basis of "Blind/disabled" (*n* = 79,936, 97.42%). Those eligible for Medicaid-only most frequently were on a 1915(C) MR/DD waiver (n = 31,390, 38.35%), 1915(B) waiver (n = 15,975, 19.47%), or 1115 waiver (n = 10,417, 12.67%), with other waiver programs occurring in less than 1% of the

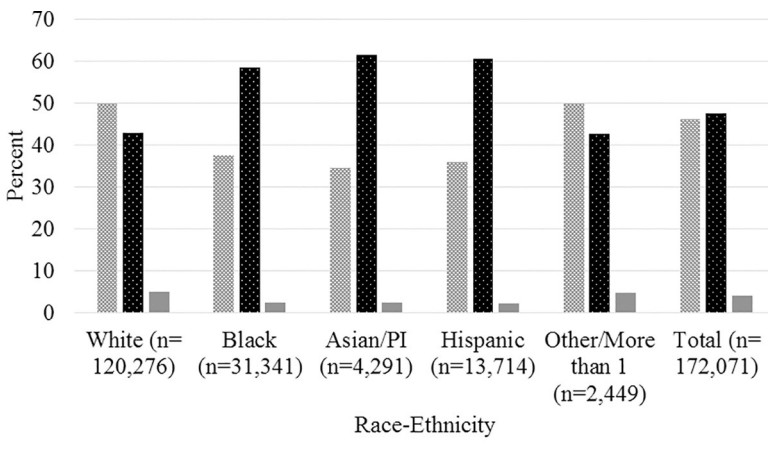

**Fig 2. Percentage of beneficiaries eligible for Medicare, Medicaid, or full-dual status by race-ethnicity among adults with autism spectrum disorder.**

Medicaid-only sample. Only 0.36% of the Medicaid-only sample (*n* = 293) were on a 1915(C) autism waiver in 2012 as an adult. Almost a quarter of the Medicaid-only beneficiaries were not eligible for a waiver overall (n = 20,677, 25.20%). Racial and ethnic differences existed in waiver program eligibility ($\chi 2(60)$ = 16000, p < .001). Significantly more Asian/Pacific Islander Medicaid-only (55.45%) and Hispanic beneficiaries (42.87%) were on 1915(C) MR/DD waivers as compared to white (37.57%), Black (35.84%), or Other/more than 1 race (34.16%) Medicaid-only beneficiaries. Asian/Pacific Islander (11.88%) and Hispanic beneficiaries (19.09%) were also less likely to 'not be enrolled in a waiver' as compared to white (25.70%), Black (28.22%), and Other/More than 1 race (29.58%) beneficiaries on Medicaid-only.

## Multivariate logistic regressions

Logistic regressions revealed that white beneficiaries with ASD were significantly more likely to be full-dual eligible compared to Black, Asian/Pacific Islander, or Hispanic beneficiaries across all age groups after controlling for gender, intellectual disability, costly chronic condition, county median income, rural county, and geographic region of residence (Models in S2 Table). Post-estimation marginal effects illustrate the predicted probability of being full-dual eligible by race-ethnicity while holding other covariates constant (Table 2).

## Spending by race and ethnicity

Among *full-dual* adult ASD beneficiaries (*n* = 79,378), a total of $6,280,520,254 was spent across both Medicare and Medicaid services in 2012 U.S. dollars. This total included the Medicare portion of payments ($5,195,809,661), the Medicaid portion of payments ($5,245,325,040), and the beneficiary out-of-pocket share of payments for Medicare and Medicaid services ($141,854,935). Among *Medicaid-only* ASD adult beneficiaries (*n* = 82,056), a total of $4,125,953,714 was spent in 2012 U.S. dollars; this included the Medicaid portion of the spending ($4,125,953,714) and the beneficiary out-of-pocket spending totaled $257,810. Among *Medicare-only* ASD adult beneficiaries (*n* = 7204), total spending was $87,586,089, which included Medicare portions of the spending ($75,407,619) and beneficiary out-of-pocket costs of $12,178,470.

**Table 2. Predicted probability of being full-dual eligible by race-ethnicity for autistic adults across the lifespan.**

|  | 18-24y [a] Model 1 | 23-34y [b] Model 2 | 35-44y [c] Model 3 | 45-54y [d] Model 4 | 55-64y [e] Model 5 |
|---|---|---|---|---|---|
|  | N = 56,753 | N = 44,282 | N = 21,534 | N = 19,493 | N = 11,418 |
| **White** | **16.97%** | **47.91%** | **77.48%** | **90.19%** | **92.45%** |
| **95%CI** | 16.57, 17.37 | 47.32, 48.50 | 76.80, 78.15 | 89.70, 90.67 | 91.90, 93.01 |
| **Black** | **14.29%** | **38.83%** | **62.33%** | **77.37%** | **80.13%** |
| **95%CI** | 13.65, 14.94 | 37.78, 39.88 | 60.77, 63.89 | 75.80, 78.93 | 77.91, 82.34 |
| **Asian/PI** | **14.15%** | **44.68%** | **75.54%** | **79.50%** | **69.85%** |
| **95%CI** | 12.38, 15.93 | 41.96, 47.39 | 71.77, 79.31 | 74.37, 84.62 | 60.60, 79.10 |
| **Hispanic** | **12.90%** | **45.06%** | **72.99%** | **88.16%** | **90.30%** |
| **95%CI** | 12.03, 13.77 | 41.42, 44.70 | 70.65, 75.32 | 85.47, 93.04 | 87.49, 93.11 |
| **Other/ More than one race** | **21.27%** | **59.63%** | **84.60%** | **89.26%** | **94.72%** |
| **95%CI** | 18.50, 24.05 | 55.65, 63.61 | 80.70, 88.50 | 86.10, 90.21 | 91.45, 97.99 |

[a-e]Full models presented in S2 Table. Models adjusted for: sex, intellectual disability, any costly chronic condition, county median income, rural county, geographic region of residence. Presented as post-estimation adjusted predicted probability of being full-dual eligible for both Medicare and Medicaid.

Data source: MMLEADS V2, 2012

**Table 3. Median annual per beneficiary spending in 2012 by race-ethnicity among autistic adults.**

| | White | Black | Asian/PI | Other/More than 1 | Hispanic |
|---|---|---|---|---|---|
| **Full-Dual Eligible** | n = 59,976 | n = 11,761 | n = 1,487 | n = 1,225 | n = 4,929 |
| **Total annual spending per beneficiary** | **$59,013** | **$47,405 (46011, 48922)** | **$42,097** | **$ 45,897** | **$46,252** |
| *Median (95%CI for 50th centile)* | **(58292, 59732)** | | **(39691, 45944)** | **(39962, 49955)** | **(44725, 47621)** |
| Medicare spending per beneficiary | $6,927 | $5,269 | $4,390 | $5,929 | $6,166 |
| *Median (95%CI for 50th centile)* | (6834, 7023) | (5068, 5451) | (4024, 4633) | (5261, 6590) | (5899, 6501) |
| Medicaid spending per beneficiary | $45,209 | $35,263 | $34,480 | $33,292 | $36,596 |
| *Median (95%CI for 50th centile)* | (44595, 45846) | (33998, 36773) | (31945, 36880) | (27904, 37080) | (34938, 37080) |
| Beneficiary out-of-pocket costs for Medicare | $388 | $310 | $260 | $365 | $338 |
| *Median (95%CI for 50th centile)* | (383, 392) | (302, 318) | (247, 280) | (336, 402) | (324, 353) |
| Beneficiary out-of-pocket costs for Medicaid | $142 | $132 | $27 | $148 | $43 |
| *Median (95%CI for 50th centile)* | (141, 143) | (128, 136) | (6, 38) | (139, 163) | (37, 48) |
| **Medicaid-only** | *n* = 51,663 | n = 18,378 | n = 2,642 | n = 1,048 | n = 8,325 |
| **Total annual spending per beneficiary** | **$29,082** | **$24,900** | **$35,821** | **$27,075** | **$29,206** |
| *Median (95%CI for 50th centile)* | **(28513, 29616)** | **(24264, 25702)** | **(34180, 37080)** | **(22451, 32344)** | **(27811, 30355)** |
| Medicaid spending per beneficiary | $29,080 | $24,893 | $35,821 | $27,075 | $29,206 |
| *Median (95%CI for 50th centile)* | (28513, 29606) | (24258, 25692) | (34180, 37080) | (22451, 32344) | (27803, 30355) |
| Beneficiary out-of-pocket costs for Medicaid | $0 | $0 | $0 | $0 | $0 |
| *Median (95%CI for 50th centile)* | (0, 0) | (0, 0) | (0, 0) | (0, 0) | (0, 0) |
| **Medicare-only** | n = 5,951 | n = 738 | n = 101 | n = 117 | n = 297 |
| **Total annual spending per beneficiary** | **$3,434** | **$1,384** | **$1,401** | **$1,621** | **$1,639** |
| *Median (95%CI for 50th centile)* | **(3250, 3738)** | **(1123, 1759)** | **(695, 2993)** | **(852, 2849)** | **(961, 2378)** |
| Medicare spending per beneficiary | $2,615 | $1,028 | $1,131 | $1,163 | $1,281 |
| *Median (95%CI for 50th centile)* | (2472, 2802) | (779, 1345) | (450, 2437) | (525, 2136) | (696, 2069) |
| Beneficiary out-of-pocket costs for Medicare | $595 | $269 | $229 | $269 | $233 |
| *Median (95%CI for 50th centile)* | (562, 632) | (223, 313) | (167, 368) | (194, 396) | (194, 289) |

Abbreviations: CI, confidence interval.

50th centile reported with binomial exact 95% confidence intervals.

Data source: MMLEADS V2, 2012.

Across these three beneficiary types (*n* = 168,638), the sum total of spending among adults aged 18 and over with ASD in 2012 was $10,494,060,057. Table 3 illustrates descriptive statistics on spending by race-ethnicity. Annual median spending per full-dual beneficiary was $14,685 less for Black, $24,515 less for Asian/Pacific Islander, $10,977 less for Hispanic, and $17,685 less for Other/More than one as compared to white beneficiaries.

## Discussion

Disparities research in autism spectrum disorders in the mid-2000s highlighted the gaps in diagnosis, services, and spending for children [1–3] but tended to ignore autistic adults. Disparity frameworks such as that posed by Kilbourne and colleagues [34] suggest that studies to address the disparities must first describe, understand, and then intervene when disparities are identified. Despite awareness of the considerable differences impacting equitable pathways to diagnosis and care for *children* with autism spectrum disorder across the past two decades, as well as attempts to understand system and provider factors influencing care, our research shows that differences in eligibility and spending on public benefits continues to negatively impact *adult* ASD beneficiaries of different race-ethnicity groups as they become eligible for services after high school for those who rely on public-sector health coverage.

Our study reveals that Black beneficiaries with ASD were significantly less likely to be full-dual eligible for both Medicare and Medicaid than white beneficiaries across all ages. Younger Asian/Pacific Islander beneficiaries (18-24y) and older Asian/Pacific Islander beneficiaries (older than 45y) were significantly less likely to be full-dual eligible than white beneficiaries. Hispanic beneficiaries were significantly less likely to be full-dual eligible than white beneficiaries only at the younger age categories (less than 45y). Those of 'Other race/More than one race', including Alaska Native and American Indian beneficiaries, were significantly more likely to be dual-eligible between 18–44 years as compared to white beneficiaries. Dual-eligibility provides beneficiaries with multiple coverage options for services. Dual-eligibility status is available to beneficiaries who both meet Social Security disability determination criteria, and are enrolled in state Medicaid programs, including waivers.

Our data suggest that full-dual beneficiaries are likely to be eligible on the basis of disability determination criteria, and not poverty, and this preliminarily suggests that pathways to dual-status on the basis of disability should consider other factors that impact eligibility for ASD individuals of different racial-ethnic groups. The pathways to dual-eligibility status are difficult to navigate for families and individuals, and those pathways differ based on a number of reasons [35]. Many individuals report that obtaining a Social Security disability determination takes more than one attempt [24]. Bilder and Mechanic noted in a study of individuals with serious mental illness that the application process for Social Security Disability Insurance (SSDI) is challenging [36]. Their results suggest that those who applied for SSDI were significantly more likely to have had doctor visits and community mental health provider visits; after controlling for severity of the mental health need, these authors concluded that the providers were a source of information about the SSI/SSDI process and facilitated the applicant's knowledge. Godtland and colleagues similarly found that external support in the SSDI process affects the outcome of disability decisions [37]. Specifically, these authors found that African-American applicants for SSA disability had similar likelihood of being awarded disability insurance as white applicants only when they were represented by an attorney at a disability hearing [37]. When African-American applicants were not represented by an attorney, they had statistically lower likelihood than white applicants of being awarded [37]. Unfortunately, in that study, significant differences in attorney representation were found, with African-American applicants being represented 58% of the time, and white applicants being represented 71% of the time [37]. Additionally, these authors found that applicants from higher wealth families were significantly more likely to be awarded SSDI. Collectively, this research points to the importance of learning about benefits from providers, understanding the rights to benefits and the process for application, and a persistence and ability to obtain assistance when needed. For families and individuals on the autism spectrum without resources of time, money, or knowledge, the systems in place appear to benefit privileged individuals, affecting both disability status and the possibility of becoming dual-eligible for health benefits. Our study points to needed policy changes to ease these disparities through education, outreach, and system navigation barriers, particularly for individuals from racial and ethnic minority groups.

Disability determination and eligibility for Medicare are often based on having a previous employment history. Autistic individuals have difficulty obtaining and maintaining employment upon entering adulthood [22, 23, 38] and it is possible that minority individuals experience disparities in both job-seeking and maintenance, thus exacerbating difficulties with obtaining disability benefits, including Medicare. These multi-system factors reflect mechanisms by which systemic racism perpetuates disparities. Solutions will need to be systemic, policy-oriented, and involve emerging and innovative access policies. For example, those implemented through the Affordable Care Act could provide bright lights for policy development. Mechanisms such as No Wrong Door for eligibility, presumptive eligibility, and

continuous eligibility would support *all* individuals, easing paperwork and administrative requirements that disproportionately impact families and individuals lacking resources or awareness.

Usually, healthcare spending is driven by co-occurring conditions; in autism, intellectual disability status is associated with nearly double the lifetime costs [32]. In our sample, Black beneficiaries were significantly more likely to have a co-occurring intellectual disability compared to white beneficiaries, but the median spending for black beneficiaries per year was 20% less for full-dual beneficiaries. Less resources were spent on the racial group with higher rates of intellectual disability, despite other literature suggesting that intellectual disability results in higher costs [39]. However, our costly chronic condition indicator suggests that perhaps Black, Asian/PI, and Hispanic beneficiaries had less incidence of an identified chronic condition when using CCW criteria. Future research needs to disentangle the contribution of co-occurring intellectual disability and other chronic conditions that affect most aging adults. Our data, because they are not actual claims, do not allow us to understand the specific contributions of these conditions to overall spending. Regardless, our multivariate regressions controlled for both intellectual disability and presence of a costly chronic condition and estimates of eligibility by race-ethnicity remained stable despite the inclusion or exclusion of these variables.

Finally, the available data from our study suggest that a single year of spending in 2012 in both Medicare and Medicaid for adults with ASD aged 18 years and older resulted in spending of close to $10.5 billion dollars across both public payer systems. Our data year of 2012 marks the age of adulthood for many children diagnosed and documented in 2002 around age 8 in the midst of growing awareness of autism spectrum disorder [40]. At that time, CDC estimated prevalence at 8-years of age to be 6.6 per 1,000 children. Subsequent CDC surveillance for 8-year old children documented in 2016 indicate a prevalence of 18.5 per 1,000 [41]. Given the increase in prevalence of 8-year old children each year since that time as measured by the CDC, our findings suggest that a large number of adults with ASD are likely to require services and supports from public payers. While not all adults will be eligible for Medicare and/or Medicaid, it is likely that spending will continue to increase as more of these children and adolescents age into adulthood. Policy makers and healthcare administrators require sustained and collaborative efforts to improve care coordination across systems, prevent mental health crisis and emergency department care, and consider the effects that lack of regular employment, safe and affordable housing, and qualified care providers will have on the future of these adults. Unless specific attention is paid to assuring equitable access to and utilization of services, racial-ethnic disparities are likely to be compounded as demand for limited public-sector resources increase.

## Limitations

Our data reflect un-duplicated beneficiary-level demographic, eligibility, and spending variables abstracted from CMS claims in 2012 into a summary analytic file, the MMLEADS v2. These beneficiaries were identified by CMS as having met criteria for autism spectrum disorder across both Medicaid and/or Medicare programs. The data have gone through multiple checks for de-duplication of spending, and available annual and monthly variables allow for researcher checks on data accuracy. One limitation of this data is that all original claims submitted to CMS are subject to billing and coding errors. Despite this, the reflected costs, whether arising from an accurate claim or not, were paid by CMS for beneficiaries meeting criteria for ASD, and therefore, from a system perspective are being used by administrators of these programs to make decisions for this condition as coded. Second, not all states reported their data for 2012 before the MMLEADS data were compiled for analysis, and some states

have incomplete or inconsistent reporting to the CMS source files as of 2011 for certain beneficiary types [42]. Therefore, our estimates of costs reflect only those states which reported their data, although any additional reporting from states without data would certainly increase the total reflected spending by Medicaid on adult ASD beneficiaries. Therefore, our estimates are likely an *underestimate of total Medicaid costs* [29] which impact estimates both for dual-eligible and Medicaid-only beneficiaries in our sample (p.102). However, for dual-enrolled beneficiaries, the demographic and enrollment variables are complete, even if state reporting of utilization and spending are not, therefore, the estimation of demographic and eligibility characteristics remain unaffected by these data anomalies. Lastly, although this MMLEADS V2 data release (2012) was the most recent available for purchase as of July 2020, the data reflect a period in U.S. during which great shifts in healthcare were occurring. It is important to use this data as a starting place for benchmarking dual-eligible and other adults on the autism spectrum, however, future researchers should consider newer MMLEADS data releases, as they become available.

## Conclusion

Public health insurance in the U.S. such as Medicare and Medicaid aim to reduce inequities in access to healthcare that might exist due to disability, income, or old age. In contrast to these ideals, our study reveals that racial-ethnic minority autistic adults who were Medicare and/or Medicaid-eligible in U.S. states in 2012 experience differences in program eligibility enrollment (single payer versus dual-eligible) and in spending compared to white beneficiaries. Most studies of autistic adults document access, utilization, and outcomes primarily among privately-insured white males. Our study is one of the first to document access and costs in a diverse, population-based cohort of publicly-insured adults, and reveals significant racial-ethnic disparities that likely affect health outcomes across the lifespan.

Peer-reviewed literature infrequently discusses systemic racism as a contributor to such disparities, primarily because it is difficult to measure and conceptualize in most existing data. While we do not have in our data specific or adequate variables that would allow us to measure the exposure individuals have had over their lifetime that perpetuate lack of access and equitable treatment, the observed differences in eligibility for Medicare and Medicaid among racial-ethnic minority beneficiaries across all age-groups, and the differences in spending across these beneficiaries (despite higher rates of intellectual disability), and our adjustment for other explanatory variables (e.g. county median income), suggest that systemic racism as a hypothesized reason should be explored in future studies. Additionally, we call for further evaluation of system supports that promote clear pathways to disability and public health insurance eligibility among those with lifelong chronic or developmental disabilities, including those with autism spectrum disorder. Policy makers and system administrators need information about this growing population of publicly-insured individuals with ASD, and our study is one of the first to document this using a very large sample from all U.S. states.

## Supporting information

**S1 Table. Chronic condition warehouse criteria for autism spectrum disorder or intellectual disability.**
(DOCX)

**S2 Table. Logistic models predicting dual-eligibility status compared to medicaid-only status.**
(DOCX)

**S1 Data. Data availability statement.**
(DOCX)

## Author Contributions

**Conceptualization:** Teal W. Benevides, Henry J. Carretta, George Rust.

**Formal analysis:** Teal W. Benevides.

**Funding acquisition:** Teal W. Benevides.

**Investigation:** Teal W. Benevides.

**Methodology:** Teal W. Benevides, Henry J. Carretta, Lindsay Shea.

**Project administration:** Teal W. Benevides.

**Software:** Teal W. Benevides.

**Supervision:** Henry J. Carretta, George Rust.

**Writing – original draft:** Teal W. Benevides, Henry J. Carretta, George Rust, Lindsay Shea.

**Writing – review & editing:** Teal W. Benevides, Henry J. Carretta, George Rust, Lindsay Shea.

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
