## [Decision Letter · Decision Letter 0]

3 Mar 2021

PONE-D-21-01256

Racial and ethnic disparities in benefits eligibility and spending among adults on the autism spectrum: A cohort study using the Medicare Medicaid Linked Enrollee Data Source

PLOS ONE

Dear Dr. Benevides,

Thank you for submitting your manuscript to PLOS ONE. After careful consideration, we feel that it has merit but does not fully meet PLOS ONE’s publication criteria as it currently stands. Therefore, we invite you to submit a revised version of the manuscript that addresses the points raised during the review process.

We look forward to receiving your revised manuscript.

Kind regards,

Kevin Lu, PhD

Academic Editor

PLOS ONE

Journal Requirements:

3) We note that you have indicated that data from this study are available upon request. PLOS only allows data to be available upon request if there are legal or ethical restrictions on sharing data publicly. For information on unacceptable data access restrictions, please see http://journals.plos.org/plosone/s/data-availability#loc-unacceptable-data-access-restrictions.

Reviewers' comments:

Reviewer's Responses to Questions

**Comments to the Author**

1. Is the manuscript technically sound, and do the data support the conclusions?

Reviewer #1: Yes

2. Has the statistical analysis been performed appropriately and rigorously? 

Reviewer #1: Yes

3. Have the authors made all data underlying the findings in their manuscript fully available?

Reviewer #1: Yes

4. Is the manuscript presented in an intelligible fashion and written in standard English?

Reviewer #1: Yes

5. Review Comments to the Author

Reviewer #1: The study described a significant racial and ethnic disparities of Medicare and Medicaid benefits eligibility among adult autism spectrum disorder (ASD). The findings also show that the total medical cost of patients with ASD existed racial and ethnic disparities.

The study limitations have been well thought of and it will be good to investigate the racial and ethnic disparities further in a population cohort that is covered by other healthcare insurances. This study will provide additional resources for further understanding and evaluation of the influence of the health outcomes from such disparities of patients with ASD for the health care policymakers.

Questions/concerns:

1) In the Methods, the authors report only 2012 MMLEADS V2 data was used for the study. Why the authors did not use more recently years, such as 2016, 2017? And is there more background information about this data?

2) Did all participants were continually enrolled in Medicaid, Medicare, or dually all year?

3) Did authors consider those participants with Medicare Part C?

4) In data analysis, the author mentioned that several states (Alaska, Colorado, Hawaii, etc.) were excluded in all models because of large standard errors. I think the potential reason might because of racial and ethnic disparities of ASD diagnosis in different states. (see this article: https://stacks.cdc.gov/view/cdc/22182). Therefore, the authors may include all states and categorize them into “Northeast, South, Midwest, and west”.

5) Living in urban or rural areas also should be considered.

6) Medical conditions should be considered as an important covariate in logistic regression model.

7) Was the cost overall cost? I think the authors should add a short paragraph in the method part to carefully describe how to calculate the cost. In addition, if it is possible based on the data available, the authors may specifically analyze for different types of the cost, such as ASD direct or indirect cost, Emergency department cost, hospitalization cost, ect., and see if it has any racial and ethnic disparities.

6. PLOS authors have the option to publish the peer review history of their article (what does this mean?). If published, this will include your full peer review and any attached files.

Reviewer #1: No

---

## [Author Response · Author response to Decision Letter 0]

12 Apr 2021

Dear Dr. Lu:

Thank you for the recent review and request for revision of our manuscript “Racial and ethnic disparities in benefits eligibility and spending among adults on the autism spectrum: A cohort study using the Medicare Medicaid Linked Enrollee Data Source” (PONE-D-21-01256). We are grateful for the supportive and helpful comments of the editor and reviewer. We hope this can be considered for the special issue on Health Services.

Upon review, we were asked to fully explain why we indicate that the data are not available: “If there are ethical or legal restrictions on sharing a de-identified data set, please explain them in detail (e.g., data contain potentially identifying or sensitive patient information) and who has imposed them (e.g., an ethics committee). Please also provide contact information for a data access committee, ethics committee, or other institutional body to which data requests may be sent.” 

Please note this information was included upon first submission, as requested, as Supplement S3 for all readers. We aimed to ensure it is clear, and provide it here for your review: 

The data used and described in this study are not publicly-available and the authors legally cannot make this data available due to a Data Use Agreement with Centers for Medicare and Medicaid (CMS) that prohibits data sharing (DUA#: RSCH-2020-55304). The data used in this study are available for purchase from CMS following a data use request from the third-party vendor, the Research Data Assistance Center (ResDAC). Researchers seeking to purchase data should visit www.resdac.org for instructions, guidance, and costs of CMS data. The specific CMS data used in this study were extracted by a third-party vendor from the Medicare-Medicaid Enrolled Linked Data Source, 2012 files. We requested a cohort-specific data purchase of this research-identifiable data using a finder file request that included the following: 

All beneficiaries in the 2009, 2010, 2011, or 2012 Condition files that were flagged with the CCW code for "Autism Spectrum Disorders", "Intellectual Disability and Related Conditions" and "Learning Disabilities and Other Developmental Delays" in the following variables: 

AUTISM_COMBINED (values=1 or 3)

AUTISM_MEDICAID (values=1 or 3)

AUTISM_MEDICARE (values=1 or 3)

INTDIS_COMBINED (values=1 or 3)

INTDIS_MEDICAID (values=1 or 3)

INTDIS_MEDICARE (values=1 or 3)

LEADIS_COMBINED (values=1 or 3)

LEADIS_MEDICAID (values=1 or 3)

LEADIS_MEDICARE (values=1 or 3)

Best Regards,

Teal W. Benevides, PhD, MS, OTR/L

Associate Professor, Department of Occupational Therapy

College of Allied Health Sciences, Augusta University

tbenevides@augusta.edu

On behalf of co-authors: Henry J. Carretta, PhD, MPH; George Rust, MD, PhD; and Lindsay Shea, DrPH, MS

---

## [Decision Letter · Decision Letter 1]

26 Apr 2021

Racial and ethnic disparities in benefits eligibility and spending among adults on the autism spectrum: A cohort study using the Medicare Medicaid Linked Enrollee Data Source

PONE-D-21-01256R1

Dear Dr. Benevides,

We’re pleased to inform you that your manuscript has been judged scientifically suitable for publication and will be formally accepted for publication once it meets all outstanding technical requirements.

Kind regards,

Kevin Lu, PhD

Academic Editor

PLOS ONE

---

## [Editor Report · Acceptance letter]

11 May 2021

PONE-D-21-01256R1 

Racial and ethnic disparities in benefits eligibility and spending among adults on the autism spectrum: A cohort study using the Medicare Medicaid Linked Enrollees Analytic Data Source 

Dear Dr. Benevides:

I'm pleased to inform you that your manuscript has been deemed suitable for publication in PLOS ONE. Congratulations! Your manuscript is now with our production department. 

Kind regards, 

on behalf of

Professor Kevin Lu 

Academic Editor

PLOS ONE